# Blazar Jets as Possible Sources of Ultra-High Energy Photons: A Short Review

Gopal Bhatta

Institute of Nuclear Physics Polish Academy of Sciences, 31342 Kraków, Poland; gopal.bhatta@ifj.edu.pl

**Abstract:** In this paper, I present a qualitative discussion on the prospect of production of ultra-high photons in blazars. The sources are a subclass of active galactic nuclei which host supermassive black holes and fire relativistic jets into the intergalactic medium. The kpc-scale jets are believed to be dominated by Poynting flux and constitute one of the most efficient cosmic particle accelerators, that potentially are capable of accelerating the particles up to EeV energies. Recent IceCube detection of astrophysical neutrino emissions, in coincidence with the enhanced gamma-ray from Tev blazar TXS 0506 + 056, further supports hadronic models of blazar emissions in which particle acceleration processes, such as relativistic shocks, magnetic re-connection, and relativistic turbulence, could energize hadrons, e.g., protons, up to energies equivalent to billions of Lorentz factors. The ensuing photo-pionic processes may then result in gamma-rays accompanied by neutrino flux. Furthermore, the fact that blazars are the dominant source of observed TeV emission encourages search for signatures of acceleration scenarios that would lead to the creation of ultra-high-energy photons.

**Keywords:** ultra-high-energy cosmic rays; active galactic nuclei; blazars; relativistic jets; supermassive black holes; non-thermal emission





## 1. Introduction

Cosmic rays (CR), in general, represent high-energy particles that originate in astrophysical events involving extreme physical environments, such as supernovae, gamma-ray bursts and active galactic nuclei (AGN). The constituent particles of CR mostly include charged particles e.g., protons, fully ionized atomic nuclei and electrons, but they can include neutral particles (e.g., neutrinos and photons), as well as neutrons at higher energies. The energy spectrum of CR can be effectively approximated by a broken power-law of the form $dN/dE \propto E^p$, where the power-law index $p$ depends upon energy; the power-law spans over more than 13 orders of magnitude, starting from $10^7$ eV and extending above $10^{20}$ eV. Although 100 years have elapsed since the discovery of cosmic rays by Victor Hess in 1912, the sources of high-energy charged cosmic rays still elude us. This is mainly because the large scale magnetic field intervening between galaxies can deflect the path of CR significantly and hide the source locations from us. Nevertheless, with advancement in detectors and computing capabilities, we have considerably enriched our knowledge on the nature of their origin, acceleration and propagation mechanisms [1]. The topic is actively pursued and widely discussed in modern high-energy astrophysics. Cosmic rays that are at the lower end of the spectrum most likely have galactic origin, e.g., particles accelerated by the shocks of supernova remnants. However, ultra-high-energy cosmic rays (UHECR; $E > 10^{18}$ eV), most likely originate in an extra-galactic environment. During their propagation UHECR interact with the all-pervading cosmic microwave background (CMB) and lose a fraction of their energies, such that the CR flux is suppressed by the energy $4 \times 10^{19}$ eV, well-known as the Greisen–Zatsepin–Kuzmin (GZK) effect [2,3]. This can create a horizon around ~100 Mpc from us, such that super-GZK UHECRs originating from beyond this GZK horizon are less likely to reach detectors on the Earth. In general, the existence of super-GZK UHECR are explained using two classes of models: *Top-down* models attempt

to explain the origin of super-GZK UHECR as a product of the decay of more energetic exotic particles, such as the decay of super-heavy dark matter and topological defects from the early Universe [4–7]; whereas, in *bottom-up* models, particles are accelerated from lower energies up to the highest energies via various acceleration processes. Of these two models, the *top-down* models are discussed as they usually predict a large flux of UHE photons which have so far not been observed by current instruments.

UHE photons, although a small fraction of primary cosmic rays, constitute a crucial component of UHECR. The search for UHE photons is important because, while most of the UHECR, being charged particles, are deflected by the intervening magnetic fields, UHE photons follow a direct path to the Earth enabling identification of their sources [8,9]. Moreover, the creation of UHE photons is an important part of the processes involved in the acceleration and propagation of the charged particles that constitute a dominant fraction of primary UHECR. In other words, UHE photons are largely associated with other messenger particles, such as charged cosmic rays and neutrinos, and possibly gravitational waves. Therefore, the study of UHE photons contributes to ongoing experimental collaboration on multi-messenger studies of the Universe.

The study of UHECR, in general, plays a crucial role in the understanding of fundamental physical laws. It is widely considered that the standard model only represents the lower energy limit of a more fundamental theory, which ought to emerge at higher energies. The emergence of novel physics at very high energies (e.g., $>10^{14}$ GeV) is anticipated by grand unified theories of electroweak and strong interactions. It should be emphasized that the threshold energies are so high that the current generation accelerators on the Earth are unable to reach them. This is why the study of the UHECR astrophysical origin is crucial for the exploration of physical laws at high energies.

In this paper, I present a qualitative discussion suggesting that blazars with their relativistic jets pointed towards the Earth could be one of the most promising sources of UHE photons. In Section 2, some of the investigations that have focused on the search for UHE photons are highlighted. In Section 3, the possibility that blazars could be the astrophysical sources that produce UHE photons is discussed in detail. Finally, in Section 4, I present my conclusions.

## 2. Search for UHE Photons

There are a number of investigations (or experiments) that are dedicated to the search for high-energy photons mainly using ground-based high-energy telescopes. While Fermi[1], a space-based telescope, regularly detects gamma-ray emission up to 300 GeV, MAGIC[2] and HESS[3], air-shower-based ground telescopes, are sensitive up to a few tens of the TeV energy range. At higher energy, the Tibet Air Shower array reported the detection of several gamma-ray photons from the Crab Nebula with energies above 100 TeV [10]. More recently, the Large High Altitude Air Shower Observatory (LHAASO) reported detection of gamma ray photons with PeV energies emitted by galactic "PeVatrons". This detection included a photon at 1.4 PeV, which represents the highest energy photon event ever observed [11]. Particles with energies more than a few tens of PeV are probably produced in large-scale extra-galactic systems, such as active galactic nuclei, jets, radio galaxy lobes, and clusters of galaxies. Super GZK UHECRs, as they propagate through the cosmic microwave background (CMB), interact through the GZK process and produce both UHE photons, also known as GZK photons, and neutrinos via the decay of neutral and charged pions, respectively. The decay of $\pi^{\pm}$ results in so-called cosmogenic neutrinos, the detection of which is one of the primary goals of some of the prominent neutrino telescopes, such as IceCube[4] and ANITA[5]. The decay of neutral pions ($\pi^0$), on the other hand, produces GZK photons, which may carry 10% of the energy of the UHECR that produced them, making them a significant component of UHECRs. It is also suggested that the Universe could be more transparent to UHE photons compared to UHE photons, such that photons could dominate over the other nuclei at the highest energies (e.g., see [12–14]). However, it appears that the exact spectrum of the UHE photons depends upon a number of factors,

including the MWL extra-galactic background light (EBL), the initial proton fluxes, source distributions and large-scale intervening magnetic fields.

Generally, detection of UHE photons on the Earth relies on methods that discriminate hadronic-induced air-shower events from those initiated by UHE photons. Some of the known properties relevant to shower evolution are utilized. An air shower initiated by a UHE photon develops more slowly compared to a hadronic cascade and, therefore, the maximum of the development (usually quantified by $X_{max}$) can occur close to the ground. The difference in the nature of electromagnetic cascades versus hadronic cascades can materialize in terms of the $X_{max}$ value, such that it is found that, on average, the simulated $X_{max}$ for photon-induced cascades is larger by 200 g/cm$^2$ compared to cascades induced by protons [15]. Moreover, the air showers initiated by UHE photons tend to show a steeper lateral distribution function, i.e., narrower in spatial extension, and show less muonic content. However, when the Landau–Pomeranchuk–Migdal (LPM) [16] effect, an effect which reduces the probability of bremsstrahlung and pair-creation processed at higher energies, is included, the ability to distinguish between UHE $\gamma$-rays and protons can be significantly weakened [17].

Although photons with energies with 1 EeV and higher have not been detected to date, there have been a number of efforts to estimate the upper limit of the UHE photon flux. Using cosmic ray observations from KASCADE and KASCADE-Grande, an upper limit of the fraction of photons with energy $3.7 \times 10^{15}$ eV to the total CR flux $1.1 \times 10^{-5}$ was estimated [18]. Similarly, EAS-MSU [19] estimated upper limits of the diffuse photon flux of $\sim$10 km$^{-2}$ sr$^{-1}$ yr$^{-2}$ between the energy range of $10^{16}$ and $3 \times 10^{17.5}$ eV. Using observations from the Pierre Auger Observatory (PAO), an upper limit of integral flux of UHE photons above $10^{18}$ eV was found to be $\sim$0.008 km$^{-2}$ sr$^{-1}$ yr$^{-1}$ at the 95% confidence level [15]. More recently, a search for UHE photons in the energies $> 2 \times 10^{17}$ eV was conducted using hybrid observations from the PAO. The study resulted in an upper limit of the integral photon flux above $10^{17}$ eV $\sim 3$ km$^{-2}$ yr$^{-1}$ sr$^{-1}$ [20].

Particle cascades initiated by UHE photons in the geomagnetic field have been studied using Monte Carlo simulations [21,22]. Similarly, propagation of UHE photons in the solar magnetosphere was also studied using Monte Carlo simulations by [23–26]. The results showed that the photon disintegrates, giving rise to extended cascades comprising of thousands of spatially correlated secondary particles which lose energy through synchrotron emission. On the Earth, the observational signature of such extended cascades can be searched in the form of temporal clustering of cosmic ray events [27]. In addition, a global collaboration named The Cosmic Ray Extremely Distributed Observatory (CREDO) [28] is dedicated to searching the footprints of such spatially correlated cosmic ray events that might have been initiated by UHE photons.

## 3. Blazar Jets Possible Source of UHE Photons

AGN powered by supermassive black holes represent some of the largest energy reservoirs in the Universe. Energy extracted from a black hole can launch powerful relativistic jets, which stream particles with speeds comparable to the speed of light. Blazars are a subclass of AGN that feature relativistic jets aligned close to the line of sight [29]. As a result, the emission is significantly Doppler-boosted and rapidly variable. These sources are characterized by high luminosity, broadband emission and variability in all temporal and spatial frequencies. The broadband emission from blazars can be detected across the entire observable electromagnetic spectrum, from radio to TeV $\gamma$-rays. Blazars can be further classified into flat-spectrum radio quasars (FSRQs) and BL Lac objects based on the presence or absence of emission lines, respectively. The broad-band spectral energy distributions can be recognized by two hump-like, non-thermal emission components [30]. The low-frequency component lying between radio and X-rays is widely accepted as being due to synchrotron emission by relativistic electrons in the jet; the high-frequency component peaking between X-rays to $\gamma$-rays can be ascribed to inverse Compton scattering of low-energy target photons [31]. The sources are also sub-divided into low-synchrotron-

peaked, intermediate-synchrotron-peaked and high-synchrotron-peaked, based on the location of the peak of the synchrotron component ($\nu_{sy}$) , that is, $\nu_{sy}$, $\nu_{sy} < 10^{14}$ Hz, $10^{14}$ Hz $< \nu_{sy} < 10^{15}$ Hz and $\nu_{sy} > 10^{15}$ Hz, respectively [32].

Blazar jets fueled by $\sim 10^9$ M$_\odot$ black holes are widely considered as candidate sources that are capable of accelerating particles up to the UHE range [33,34]. It is highly likely that CRs with energies about $10^{20}$ eV, nearly seven orders of magnitude more than the energy of the particles produced at the Earth accelerators, originated from sources that belong to the AGN family. Radio-loud AGN with their large scale (kpc/Mpc) jets, satisfying the Hillas criterion $E_{max} = qBR$ that would be required to be fulfilled by potential acceleration sites, provide the most promising venues that are favorable to particle acceleration up to EeV energies. Blazars are the dominant discrete sources that contribute the TeV gamma-ray emission observed from the Earth. Moreover, the high-energy neutrino event IC 170922A was found to coincide with an enhanced $\gamma$-ray emission from the TeV blazar TXS 0506 + 056 [35,36]. The coincidence bolstered the idea that blazars host conditions favorable to the acceleration of particles that are required to produce PeV neutrinos and, therefore, are the most probable source class associated with discrete neutrino events. Although the exact production mechanism of neutrinos in blazars is still debated, the particles are closely linked to the production of UHECRs. Several possible scenarios leading to neutrino emission from blazars have been extensively discussed in several papers (e.g., see [37–41]). Highly accelerated protons, on interaction with the ambient medium, give rise to $pp$ interaction leading to neutrino production [42]. Interaction of high-energy protons with the internal and external photon fields might also result in the photo-production of pions, which subsequently decay to neutrino emission [43,44]

Following the bottom-up scenario, UHE photons can be produced in blazar jets in the course of particle acceleration of CRs up to very high energies. Therefore, they are most likely to originate near the regions that are co-spatial to the population of UHE charged particles. The UHE protons (or in general charged UHECR) lose energy via mainly proton-proton (pp; $p + p \rightarrow \pi^{0/\pm} + X$) and proton-photon ($p\gamma$) collisions. These interactions result in charged or neutral pions which further decay, producing neutrinos and gamma-rays, respectively. Furthermore, a $\gtrsim E \times 10^{19.5}$ eV proton, on interaction with the photon background, generates a pion, which rapidly decays into two photons or an electron/muon and neutrino depending on its charge ($p + \gamma \rightarrow p/n + \pi^{0/+}$); whereas for UHE protons with energies $\lesssim E \times 10^{19.5}$ eV, the photopion interaction yields an electron-positron pair ($p + \gamma \rightarrow p + e^+ + e^-$) which consequently loses energy via synchrotron emission. In addition, UHECR can induce cascade radiation in the extragalactic background light, resulting in a high-energy photon field with energies as high as 10 EeV. The signatures of such processes can be observed in the spectra of luminous blazars [45]. Photomeson interaction of neutrons outside the blob can also produce UHE photons which, along with neutrons and neutrinos, can escape the BLR to form a well-collimated neutral beam of the jet. It is also suggested that protons can be accelerated to the UHE range close to the inner jet region such that these UHECR, in turn, power a neutral beam of neutrinos, neutrons, and $\gamma$-rays from p-$\gamma$ photopion production [46]. In fact, the transportation of energy via beams of energetic neutral particles, e.g., neutrons and photon beams, could be one means of powering the hot spots and lobes visible in radio and X-ray frequencies [41]. However, in some cases, the production of an excess of neutrinos via photo-pionic processes may require large target photon densities, making the source opaque to high-energy gamma-rays [47]. This might create some doubt that the two different messengers could originate from the same location.

The production of UHE photons is linked to the particle acceleration mechanism that can accelerate the charged particles up to EeV range; of the several particle acceleration mechanisms possible in the relativist jets of radio-loud AGN, some of the most widely discussed scenarios are based on relativistic shocks, magnetic re-connection and turbulence. In weakly magnetized jets, relativistic shocks are the dominant mechanism to accelerate the particles to very high energies [48]. At the shock front, following Fermi acceleration, the

particles gain an average energy of $\langle \Delta E / E \rangle = 4/3(v/c)$ per crossing, as they frequently cross the shock wave front back and forth. In the finite shock scenario, the maximum energy attained by a proton in a finite shock with speed $\beta_s$ can be given as

$$E_{p,max} \simeq 7.8 \times 10^{20} \beta_s (B/G)(r_\perp/pc)eV,$$

and it is easy to see that, as the shock travels a pc scale distance in the blazar jet with a typical magnetic field 1 G, the proton is accelerated up to energies of tens of EeV (see [49]).

However, it may not be possible for a single shot of relativistic shocks in the radio jets to accelerate even the lighter charged particles to the UHE domain [50]. Therefore, it is suggested that lower-energy CRs, such as those produced in supernova remnants, may enter the AGN jet and be boosted to the UHE domain through multiple shock events, yielding UHECR emission that may be beamed along the jet axis [51]. An in-depth analysis exploring the nature of the variability of gamma-ray emission from a sample of blazars using decade-long Fermi/LAT (100 MeV–300 GeV) observations showed that the power spectrum density (PDS) slope indexes of the variable gamma-ray emission were found to be ∼1, and the flux histograms were best described using a lognormal probability distribution function (PDF) [52]. The PSD with slope index ∼1, well-known in flicker noise, could imply that the variability is generated by a long-memory process, with the lognormal PDF often interpreted as a multiplicative process. Such a result provides evidence that particles emitting gamma-rays retain imprints over many orders of temporal and flux scales. Together, this supports the idea that it might take the entire spatial extent of the jets for the particles to gain very high energies.

In the case of highly magnetized jets, the conditions are favorable for magnetic reconnection events [53]. Such events take place when two oppositely polarized magnetic regions in conduction plasma collide in the jets; consequently energy is released as magnetic lines which partially break and rearrange to form a stable configuration. Magnetic reconnection events, rampant along the jets, can accelerate the particle up to very high energies. Compared to a gradual acceleration at the shock-wave front the magnetic reconnections are sudden and rapid in nature [54]. The minute timescale variability in TeV emission, as observed by the HESS, supports such fast outburst emission from extremely compact regions moving with Lorentz factors that are much larger than the typical bulk Lorentz factors 10–20 [55]. Magnetic re-connection events might, in some cases, also lead to the formation of mini-jets within the main bulk jet. This has the net effect of boosting the energy of the particles, including photons, by Doppler factors according to $\Gamma = \Gamma_1 \Gamma_2 (1 + v_1 v_2 cos\theta)$, where $\Gamma$ is the resultant Lorentz factor of the particles in the mini-jet that is ejected with $\Gamma_2$ along an angle $\theta$, with the main jet propagating with a $\Gamma_1$ bulk Lorentz factor [56].

In blazar jets a population of photons might also undergo multiple inverse-Compton scattering by co-spatial UHECRs, thus gradually raising the energy of the photons to very high energies. A UHE photon (after some optical length) might later disintegrate in the locally strong magnetic field, by a pre-shower effect, resulting in an extended shower of particles. This population of secondary particles with lower energy can lose energy by synchroton emission creating blazar flares visible over multiple wavebands. Therefore, the flares that appear (quasi-)simultaneously in several wave bands could be evidence that they are likely to result from the disintegration of an ensemble of high-energy photons. A strong correlation observed in [57] might be an observational signature of such processes. Similarly, in a turbulent jet scenario, particles can be stochastically accelerated along the jet (e.g., see [58]), as magnetized cells move randomly, mimicking the magnetic mirrors, as in the second order Fermi acceleration scenario [59,60].

## 4. Conclusions

The search for UHE photons of astrophysical origin is an actively pursued area of research in the field of high-energy astrophysics. The search for UHE photons is also a part of the multi-messenger approach to the study of violent astrophysical events in the

Universe. These studies provide insights into some of the most exciting topics, including neutron stars, binary black holes, AGN jets and large-scale structures, such as galaxy clusters. UHE photons mainly result from the decay of neutral pions originating in hadronic interactions involving UHECR. In such a context, it is natural to consider blazars as the most promising candidate source class to produce UHE photons. The kpc-scale relativistic jets of blazars host some of the most violent particle acceleration processes, such as relativistic shocks, magnetic reconnection and turbulence, making them highly efficient cosmic particle accelerators. Long memory processes, log-normal flux distribution of gamma-ray flux, and correlations among MWL emissions from blazars can potentially provide observational signatures of the conditions in which UHE photons can originate. More quantitative treatment of the subject will follow in future work.

**Funding:** I acknowledge financial support by the Narodowe Centrum Nauki (NCN) grant UMO-2017/26/D/ST9/01178.

**Data Availability Statement:** Not applicable.

**Acknowledgments:** I would like to thank the anonymous referees for their constructive comments and suggestions that greatly improved the manuscript.

**Conflicts of Interest:** The author declares no conflict of interest.

## Abbreviations

The following abbreviations are used in this manuscript:

| | |
|---|---|
| AGN | Active galactic nuclei |
| BLR | Broad-line region |
| CR | Cosmic rays |
| CREDO | The Cosmic Ray Extremely Distributed Observatory |
| EBL | Extra-galactic background light |
| HESS | High Energy Stereoscopic System |
| LPM | Landau–Pomeranchuk–Migdal |
| MAGIC | Major Atmospheric Gamma Imaging Cherenkov Telescope |
| MC | Monte Carlo |
| PDF | Probability density function |
| PSD | Power spectral density |
| UHE | Ultra-high energy |
| UHECR | Ultra-high-energy cosmic rays |

## Notes

1   https://fermi.gsfc.nasa.gov/ (accessed on 30 August 2022).
2   https://www.mpp.mpg.de/en/research/astroparticle-physics-and-cosmology/magic-and-cta-gamma-ray-telescopes/magic (accessed on 30 August 2022).
3   https://www.mpi-hd.mpg.de/hfm/HESS/ (accessed on 30 August 2022).
4   https://icecube.wisc.edu/ (accessed on 30 August 2022).
5   https://www.phys.hawaii.edu/~anita/ (accessed on 30 August 2022).

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
