# Peer review of "Blazar Jets as Possible Sources of Ultra-High Energy Photons: A Short Review"

_universe, doi:10.3390/universe8100513_

Round 1
Reviewer 1 Report
Dear Authors,
I red your paper entitled "Blazar jets as possible sources of ultra-high energy photons" with an interest. The paper is devoted to a discussion of possibility of particles' acceleration in blazars' jets to very high energy.
While I found the paper to be well written, in my opinion it presents only well known results which are discussed in the literature for a several decades.
Still I believe that the work could be published as a short review on the topic. I strongly suggest authors to change the title and include a word "review" into it, to stress the review context of the paper. The authors may want also to modify the abstract accordingly.
Minor corrections:
1. Please check the text for grammar/repetitions (e.g. line 18 -- of of)
2. Line 30 -- The cosmic rays that are at the lower end of spectrum (e.g. $E> 10^{15}$ eV) ... -- at line 22 authors mentioned that the CR spectrum starts from 10^7 eV. Thus I would avoid to call 10^{15} eV "the lower end of spectrum"
Author Response
Dear referee,
Please find the response to the referee report as attached.

Reviewer 2 Report
The manuscript reviews the idea that high-energy blazar jets may be the primary source of the highest energy astrophysical gamma-rays.
The manuscript is mainly a literature review, bringing together numerous articles which have been previously published into a single document. The article itself does not qualitatively synthesize any of the results from the literature; it simply tries to connect the dots, but does not adequately prove that any of the linkages have any validity.
There are some serious gaps in the literature used in this paper. For example, the idea that simultaneous neutrino and gamma-ray emission from TXS 0506 might result from a single pion origin, as promoted by this paper, is seriously challenged by Halzen et al. (2019), who finds that the target dept necessary for sufficient production of the observed neutrino flux, would be highly opaque to high energy gamma-rays. This raises serious problems with the idea that the two different messengers might originate from the same location in TXS 0506.
Similarly, there is a discussion in the paper about the use of the average X_Max to discriminate between gamma-ray induced EAS and hadronic (proton) induced EAS. The presented argument is rather simplistic and ignores previous literature work to demonstrate the LPM effect at higher energies. These results show reduced usability of X_Max to discriminate between protons and gamma-rays.
There are a large number of English language mistakes in the document. I've reviewed the manuscript and provided extensive comments in the attached handwritten notes.
It is my opinion that the manuscript, in its present form, provides only scant, qualitative original content on this topic compared to what already exists more quantitatively in the literature.

Author Response
Dear referee,
Please find the response to the referee report attached.

Round 2
Reviewer 2 Report
The author has made significant improvements to address previous concerns. There are still a few small typos (see attached) and also one statement on page 1 which needs to be revised, otherwise it would be an incorrect statement (i.e. neutrinos and photons can be observed at both lower and higher energies, but neutrons will not be observed (due to their decay time) except at the highest energies).
As a short introductory review of the linkage between UHE gammas and CR production, the paper is useful for publication. The paper itself does not advance a new theory or result, which limits the broader significance of the content.

Author Response
Please find the response to the referee's comments on the first revision as attached.
